# Assessment of the Graft Quality and Patency during and after Coronary Artery Bypass Grafting

**DOI:** 10.3390/diagnostics13111891

**Published:** 2023-05-29

**Authors:** Matiullah Masroor, Ashfaq Ahmad, Yixuan Wang, Nianguo Dong

**Affiliations:** 1Department of Cardiovascular Surgery, Union Hospital, Tongji Medical College, Huazhong University of Science and Technology, Wuhan 430022, China; dr.matiullahmasroor1@gmail.com (M.M.);; 2Department of Cardiothoracic and Vascular Surgery, Amiri Medical Complex, Qargha Rd., Kabul 1010, Afghanistan; 3Department of Ultrasound Medicine, Union Hospital, Tongji Medical College, Huazhong University of Science and Technology, Wuhan 430022, China; ashfaqtajjak@gmail.com

**Keywords:** coronary artery bypass surgery, revascularization, graft quality assessment, angiography, CTA

## Abstract

Coronary artery bypass grafting (CABG) is the gold standard procedure for multi vessels and left main coronary artery disease. The prognosis and survival outcomes of CABG surgery are highly dependent on the patency of the bypass graft. Early graft failure which can occur during or soon after CABG remains a significant issue, with reported incidences of 3–10%. Graft failure can lead to refractory angina, myocardial ischemia, arrhythmias, low cardiac output, and fatal cardiac failure, emphasizing the importance of ensuring graft patency during and after surgery to prevent such complications. Technical errors during anastomosis are among the leading causes of early graft failure. To address this issue, various modalities and techniques have been developed to evaluate graft patency during and after CABG surgery. These modalities aim to assess the quality and integrity of the graft, thus enabling surgeons to identify and address any issues before they lead to significant complications. In this review article, we aim to discuss the strengths and limitations of all available techniques and modalities, with the goal to identify the best modality for evaluating graft patency during and after CABG surgery.

## 1. Background

Despite the advancement in percutaneous coronary intervention (PCI) technology, CABG surgery is still the gold standard procedure for left main and complex multi vessels coronary artery disease [1]. Due to the fact that PCI is less invasive and that new technology has improved its results over time [2], there are some initiatives to make CABG less invasive while achieving the same clinical outcomes [3,4,5]. Off-pump coronary artery bypass grafting (OPCAB) is the first step towards less invasiveness followed by minimal invasive direct coronary artery bypass grafting (MIDCAB), hybrid coronary revascularization (HCR), robotic-assisted coronary artery bypass grafting (RACAB), and total endoscopic coronary artery bypass grafting (TECAB) [5]. Numerous studies have shown that CABG is effective as long as the grafts are patent. The PREVENT-IV trial resulted in 13.9% and 0.9% mortality or myocardial infarction in patients with- and without saphenous vein graft failure, respectively, at 1-year follow-up on angiography [6]. Graft failure is a troublesome event and leads to several complications including myocardial ischemia, refractory angina, arrhythmias, low cardiac output, and fatal cardiac failure. Nearly 3–10% of graft failure occurs immediately, during, or soon after surgery, and technical errors play a leading role [7,8,9,10,11]. About 2–9% of the internal mammary artery (IMA) and nearly 2–20% of saphenous vein grafts fail one year after surgery [12,13,14,15]. Intraoperative harvesting techniques and preparation of IMA have been discussed in the literature which can improve the quality of graft [16]. One of the techniques to improve the outcomes of CABG is the use of quality control tools during surgery to assess the quality and patency of the graft and to revise it if necessary [17]. Multiple new non-invasive modalities for graft assessment during and after surgery have been used to date, but the best is yet to be known. In the beginning, the grafts were assessed by electromagnetic flowmeter [18], thermal coronary angiography [19], doppler flow analysis [20,21], etc., which have not recently been used and have been replaced by modern techniques. In this review article, we describe every potential method of graft assessment during and after surgery to determine which method is most effective and can be safely used clinically.

## 2. Intraoperative Graft Patency Assessment

Conventionally, the quality and flow of the graft were checked with naked eyes before anastomosis, as well as feeling the flow in the graft with digital palpation, checking the patient’s hemodynamic condition, new ECG changes for ischemia, and new heart wall motion abnormalities with echocardiography after anastomosis [22]. Unfortunately, it is hard to detect graft failure by visual checking, classical monitoring, and palpating the graft. A graft dysfunction can occur without hemodynamic, ECG, or echocardiography changes [10]. There are some advanced techniques to detect the occlusion and severe stenosis of the graft or anastomosis even in the absence of hemodynamic, ECG, or echocardiographic changes. The best time to determine the quality of the graft (especially in the case of graft failure) for a surgeon is during the surgery. During this time, the surgeon has the opportunity to revise the grafts which can, in turn, help the surgeon’s frustration as well as improve the patient outcome. There are very limited studies available in the literature comparing these new noninvasive intraoperative modalities to invasive coronary angiography (ICA). The available studies comparing intraoperative noninvasive graft assessment modalities with ICA are presented in Table 1.

### 2.1. Invasive Coronary Angiography (ICA)

Invasive coronary angiography is the gold standard investigation for graft quality assessment during surgery [7,27]. All newer techniques should be compared with ICA for their validity [28]. A study by Hol et al. [8] performed intraoperative angiography for 186 patients with 427 grafts. The CABG was performed on-pump, off-pump through sternotomy, and off-pump MIDCAB. Out of 427 grafts, 18 grafts (4.2%) were revised based on angiography results. The revision rate was 1.1%, 6.4%, and 6.5% for on-pump, off-pump through sternotomy, and MIDCAB groups, respectively. Twelve out of eighteen grafts had a problem in conduits while six out of eighteen had a problem in the distal anastomosis area. The authors believed intraoperative angiography could save many grafts which otherwise could have been occluded, and encouraged the use of intraoperative angiography [8]. Another study by the same group showed a 5% graft revision rate based on on-table angiography [7]. At the same time, they believed that sometimes it was hard to interpret the intraoperative angiography results because not all negative findings affected future follow-up patency. The sensitivity, specificity, positive predictive value (PPV), and negative predictive value (NPV) of intraoperative angiography, compared with follow-up angiography in their study, were 42%, 82%, 38%, and 84%, respectively [7]. Izzat et al. reported an 8% graft revision rate based on intraoperative angiography results [10]. The unavailability of an angiography machine in the operating room (OR) in most hospitals is one of the challenges for its routine use, and the availability of more hybrid operating rooms in hospitals will address this issue. A hybrid OR serves as a complete OR as well as a complete catheterization laboratory. It allows surgeons to perform angiography on completion of surgery and detect abnormal grafts. It also offers the opportunity to solve the problem with either PCI or surgical Intervention before the patient leaves the OR [27].

### 2.2. Intraoperative Fluorescence Imaging (IFI)/Indocyanine Green (ICG) Angiography

Intraoperative fluorescence imaging is an intraoperative invasive angiography-like modality but uses fluorescent indocyanine green (ICG) dye. This is less invasive compared with catheter-based angiography. The dye is injected through the central vein and images of grafts, anastomosis, and native coronary vessels are achieved. ICG dye has been in clinical use for many decades but its use in quality assessment in CABG surgery started at the beginning of the 21st century [29,30,31]. Desai et al. performed a study of 120 patients with 348 grafts who performed ICG angiography intraoperatively. They found that 5 out of 120 patients (4.2%) had major graft problems which needed revision of anastomosis or construction of new grafts. Six patients with twenty-two grafts underwent X-ray angiography as a pilot study for a future randomized clinical trial. The sensitivity, specificity, PPV, and NPV of ICG compared with conventional angiography for more than 50% graft stenosis were 100% each. ICG detected 3 grafts to be dysfunctional and 19 grafts patent, and conventional angiography gave the same results. The authors concluded that ICG angiography could detect patients who needed graft revision and would otherwise have gone unnoticed [24]. Another study by Waseda et al. compared intraoperative transient time flow measurement (TTFM) with the IFI system. They analyzed 137 patients with 507 grafts with ICG angiography and found 21 grafts with unsatisfactory TTFM results to be acceptable by ICG angiography. At the same time, six grafts with acceptable TTFM results were considered graft failure by ICG angiography which needed immediate revision. The authors believed the IFI system enabled on-site assessment of the grafts and anastomosis and provided functional and morphological information [32]. Another OPCAB study by Oliver et al. included 38 patients with 124 grafts. Out of 124 grafts, 107 were analyzed. Out of 107 grafts, 4 grafts needed revision. Three grafts had anastomotic stenosis and one graft had conduit dissection. The authors believed that ICG images were equivalent to angiography without catheter insertion. Additionally, ICG images could help to show the course of coronaries that would be difficult to find in obese patients. They believed ICG-based imaging technology was feasible and easy to use for the assessment of graft quality and patency [33]. Desai et al. in a randomized control trial, compared ICG with conventional angiography in 46 patients with 139 grafts. There were two false-negative and no false-positive results on the ICG angiogram. Compared with a conventional angiogram, an ICG angiogram had 100% specificity and 83% sensitivity. They encouraged the use of ICG angiography during CABG surgery [23].

### 2.3. Transient Time Flow Measurement (TTFM)

Even though coronary angiography is the gold standard and most reliable method for intraoperative graft patency, when considering the cost, availability, additional surgery time, and risk, it may not be a very practical and feasible way of assessing graft patency. Assessing the patency of the graft through a transient time ultrasound is a noninvasive, simple, and reliable method of flow measurement of the graft while the patient is still in the OR [34]. TTFM was first used for quality control in CABG in 1998 by Walpoth et al. and has been constantly in use since [35]. The 2018 ESC/EACTS guidelines on myocardial revascularization recommend the use of TTFM for intraoperative quality assessment [36]. TTFM can detect technical errors and provide the opportunity to correct the problem during surgery. The three main parameters measured during TTFM and its recommended values are mean graft flow (MGF) of >15 mL/min, pulsatility index (PI) of less than 5, and diastolic filling (DF) of greater than 50% [22,37,38]. A revision is recommended if two of the above three criteria are not met. A study by Walker et al. [25] compared TTFM with intra or postoperative angiography in 160 LITA-LAD grafts. The proportion of FitzGibbon type A grafts was 152/160 (95%), and 8 grafts were defective with 3 being type B and 5 being type O grafts. Based on the above given parameters of TTFM, no graft would have been identified as defective. According to the FitzGibbon grading, a graft with normal flow or less than 50% reduction in diameter is type A, a graft with greater than 50% stenosis or reduction in diameter is type B, and a graft without flow that is considered to be occluded is type O [39]. In the study by Walker et al. out of three parameters measured by TTFM, only MGF was significantly different between patent and defective grafts with a mean flow of 34.3 ± 16.8 mL/min and 23.9 ± 12.5 mL/min, respectively. Considering the above values as a standard, their study would have predicted six false positives based on MGF, one false positive based on PI, and two false positives based on DF. They believed that high flow and lower PI were not a guarantee against graft dysfunction [25]. A study by D’Ancona and colleagues for 161 patients with 323 distal anastomoses who underwent OPCAB where all patients’ bypass grafts were evaluated using TTFM intraoperatively, found that 32 grafts (9.9%) needed revision based on the results obtained by TTFM. The decision to either accept the graft or revise the graft was based on the flow curves and PI or both. All the revised grafts were found to have technical errors such as thrombus, kinking of the graft, intimal flap, or dissection. They found the TTFM very helpful and strongly recommended the use of TTFM in both off-pump and on-pump CABG surgeries [11]. Normally the mean flow in the sequential grafts is higher than the individual graft [40]. A study by Yang et al. introduced a new TTFM method which they named flow reduction TTFM for sequential grafting [41]. In sequential grafting, because of the high flow in the graft, less than critical anastomotic defects might not significantly decrease the flow and, therefore, might be missed by conventional TTFM. They compared the conventional and new (flow reduction) TTFM methods for all sequential anastomoses. In the conventional method, the probe would be placed 2 cm proximal to the target anastomosis with normal graft flow, while in the new method, a bulldog clamp was applied a few centimeters distal to target anastomosis to reduce the flow, and then the flow was measured the same way as the conventional method. Two distal anastomoses in the middle of the sequential grafts were found defective on flow reduction TTFM, which were missed by conventional TTFM and were revised subsequently. They believed that the temporary flow reduction method increased the sensitivity of TTFM for less than critical anastomotic defects of sequential grafting [41]. A randomized trial (GRIIP) compared the imaging group (revision based on TTFM and IFI) and control group (revision based on surgeon judgment and conventional approach) with 78 patients in each group and the same number of grafts in both groups. The major adverse cardiovascular events (MACE) (MI, repeat revascularization, and death) were similar between the groups (7.7%). After one year, angiography was performed for 55 patients with 160 grafts, and 52 patients with 152 grafts, in imaging and control groups, respectively. Single or multiple graft occlusion was comparable between the groups (30.9% imaging group) and (28.9% control group). They believed the use of TTFM and IFI was safe but did not lead to a reduction in graft occlusion at 1-year follow-up [22]. A study by Leviner et al. suggested different cut-off values of TTFM for on-pump vs. off-pump CABG [42]. An RCT comparing TTFM to angiography showed nine grafts’ TTFM values to be normal but had greater than 50% stenosis on the angiogram. Meanwhile, two grafts had abnormal TTFM values which were normal on the angiogram [23]. The limitation of TTFM is its lower sensitivity to detect greater than 50% occlusion of the graft [23,25,43]. One limitation of the TTFM was its inability to detect technical errors distal to the anastomosis. Normal TTFM values were shown in the case of distal LIMA-LAD anastomosis occlusion because of a technical error with the preserved retrograde flow in LAD and very limited or no antegrade flow in LAD distal to the anastomosis [23].

### 2.4. Doppler Ultrasonography

#### 2.4.1. High-Frequency Epicardial Echocardiography (HEE)

High-frequency epicardial echocardiography (HEE) was developed for the assessment of morphological and functional information of the graft and anastomosis quality. Several clinical reports have established its efficacy intraoperatively [26,44,45,46,47]. Anastomosis assessment is performed by using a high-frequency linear probe, with a frequency ranging from 10 to 13 MHz. A study by Suematsu et al. used HEE and power Doppler imaging for the assessment of coronary arteries and graft anastomoses during CABG. The maximal luminal diameter of the graft at the site of anastomosis was obtained intraoperatively by using power Doppler imaging and was compared with a postoperative coronary angiogram. Excellent correlation was observed between these modalities and they concluded that the patency of anastomosed graft could quickly be evaluated by epicardial echocardiography [44]. Similarly, a study by Budde et al. evaluated three different types of intentionally created coronary anastomosis construction errors (suture cross-overs, purse-string, or deep toe stitch) by using HEE with a linear probe of 13-MHz frequency. HEE enabled the detection of construction errors with high specificity and sensitivity [45].

On the contrary, Hol et al. conducted a study where they compared the measurement of graft diameters and graft quality assessment using epicardial ultrasonography with those measured using intraoperative angiography. They found poor correlations between the two methods for graft diameter. However, epicardial ultrasonography detected 13% of the abnormal grafts while ICA found 23% of the abnormal grafts. The specificity, sensitivity, PPV, and NPV for HEE compared with ICA were 90%, 22%, 40%, and 79%, respectively. The authors concluded that although epicardial ultrasonography was useful in assessing graft quality intraoperatively, angiography was superior in identifying grafts that require revision. Therefore, epicardial ultrasonography has the potential of assessing graft morphology but its ability to predict graft that needs revision should be further evaluated in comparative studies [26].

#### 2.4.2. Transesophageal Echocardiography (TEE)

Transesophageal echocardiography (TEE) was one of the accepted modalities used for the assessment of grafts intraoperatively. Several studies by Hiroshima University Hospital performed an intraoperative assessment of grafts by using TEE [48,49,50]. In 90.4% of cases, the left internal thoracic artery (LITA) was successfully visualized using TEE. The clamp-and-decamp test, when combined with TEE, allows for the evaluation of the LITA’s patency, stenosis, or the existence of a remnant branch [48]. Additionally, another study by Orihashi et al. evaluated the quality of saphenous vein or gastroepiploic artery grafts to the posterior descending artery (PDA) [49]. The grafts were successfully visualized in 95.2% of cases. Postoperative ICA results were well correlated to the intraoperative TEE results. The flow signals were easily detected in 17/20 grafts, but were hard to detect in the other 3 patients, which were determined to be occluded on postoperative angiography [49].

In a retrospective study of 51 patients by Kuroda et al. who underwent LITA-LAD grafting evaluated by TEE intraoperatively and were examined with ICA postoperatively [50], the researchers measured the flow velocity intraoperatively by TEE after anastomosis of the LITA graft. The LITA was detected in 88% of patients intraoperatively with TEE. Peak and mean velocities and velocity time integral ratios were measured and the critical values for them were 0.60, 0.73, and 1.06, respectively. The specificity for peak velocity, mean velocity, and velocity time integral ratio, was 92%, 94%, and 89%, respectively, while the sensitivity was found to be 100% for each [50]. The authors concluded that the assessment of LITA quality by TEE intraoperatively was a useful and powerful tool during CABG surgery.

## 3. Postoperative Graft Quality Assessment

It is crucial to monitor the graft after CABG surgery because life expectancy and quality of life are typically dependent on the quality of the bypass graft. According to a study, the internal mammary artery occlusion rate in the first year after surgery is 5.7% in men and 3.4% in women [51]. Saphenous vein graft shows a lower patency rate compared with ITAs. Usually, 12% of saphenous vein grafts occlude within six months after revascularization, 25% after 5 years, and the patency rate comes down to 50% at ≥15 years postoperatively [13]. The ischemic symptoms may be related to graft failure and occlusion, but 50% of anginal symptoms within five years after surgery are related to the progression of the obstruction of native coronary arteries [52]. As late survival solely depends on the patency of the graft, follow-up is mandatory and unavoidable [53].

As discussed for intraoperative graft quality assessment, we have multiple modalities to assess graft quality postoperatively as well. Different tools for assessment are invasive angiography, computed tomography (CT), magnetic resonance angiography (MRA), and transthoracic echocardiography. The available studies comparing postoperative noninvasive graft assessment modalities with ICA are listed in Table 2.

### 3.1. Invasive Coronary Angiography (ICA)

Angiography is the investigation of the standard reference for the evaluation of postoperative graft quality and patency and is routinely used for follow-up graft assessment [6,69,70,71,72]. It is the best choice to know the patency and functional assessment of the bypass grafts and native coronaries. It is also superior in being diagnostic as well as therapeutic at the same time, which is best in the case of acute myocardial infarction and cardiac arrest [73]. As a gold standard, its efficacy has been demonstrated in intraoperative assessments revealing defective grafts, which were subsequently confirmed to have issues upon revision. Therefore, the intraoperative clinical efficacy of this technique has been irrefutably established. However, due to the lack of superior investigative alternatives, the postoperative performance of ICA remains unattainable. Nevertheless, in order to verify their veracity and precision, all emerging technologies must be compared against this time-tested gold standard. ICA also has its shortcomings, such as, it being a complex and time-consuming procedure, and exposure of the bypass grafts will further prolong the procedure time. It is, therefore, exposing the patient to more radiation and a large amount of contrast [55]. Due to its invasive nature, it is associated with a potential risk of harm, such as vascular trauma, arterial dissection, arrhythmia, stroke, angina, myocardial infarction, and, rarely, embolism causing renal failure, which can further increase following CABG. These drawbacks limit its routine use as a quality control tool for graft assessment in the postoperative setting [62,74]. In stable patients, ICA carries a 0.08% risk of MI and a 0.7% risk of minor complications [75]. According to the AHA Committee on cardiovascular imaging and intervention scientific statement [76], ICA has a 0.2–0.3% risk of major adverse events (death, stroke, and MI) during or within 24 h of ICA, and the risk of minor complications (most of which are related to catheter insertion site) is 1–2%. It also has a considerable amount of discomfort because of its invasiveness. The high expense of this procedure can be attributed to the use of expensive instruments, additional time, as well as the need for a skilled team of physicians and technical experts, and resources [76].

### 3.2. Computed Tomographic Angiography (CTA)

As mentioned earlier, invasive angiography has its limitations, which have led patients to favor noninvasive procedures, and they can be easily persuaded to undergo such procedures. Healthcare workers are trying to achieve the best possible results of angiography with noninvasive techniques. CTA represents an effort in this direction, with various types of CTA available to conduct angiography to evaluate graft quality. As technology advances, more advanced machines are emerging in the market, and this journey continues to evolve. A study of symptomatic patients 10 ± 5 years after CABG surgery with 64-slice CT by Malagutti et al. showed that CTA was able to detect patency of graft with 100% sensitivity and 98.3% specificity taking ICA as a reference. They also concluded that overestimation of obstruction occurred in native coronary arteries especially when calcification was present [62]. A meta-analysis analyzed the sensitivity and specificity of 64-slice CT for graft occlusion and stenosis greater than 50% for patients presenting with angina or suspected MI symptoms after coronary artery bypass surgery. Sensitivity and specificity for the graft occlusion were 0.99 (95% CI: 0.97–1.00) and 0.99 (95% CI: 0.99–1.00), respectively, with an area under the curve (AUC) of 0.99. Sensitivity and specificity for any coronary artery bypass graft with stenosis more than 50% were 0.98 (95% CI: 0.97–0.99) and 0.98 (95% CI: 0.96–0.98), respectively, while AUC was 0.99. Neither the time from graft implantation nor age affected specificity and sensitivity on meta-regression [53]. Another meta-analysis of 15 best evidence articles with 723 patients and 2023 grafts provided the specificity, sensitivity, PPV, and NPV, of 96.7%, 97.6%, 92.7%, and 98.9%, respectively, for occluded or greater than 50% stenosed grafts [77]. According to certain authors, due to the potential complications associated with ICA, it was suggested that CTA, being a non-invasive method, should be favored instead [65,75]. With the development of each generation of CTA machines, the reliability of the detection of graft occlusion and stenosis has improved. The recent development of achieving fractional flow reserve (FFR) and perfusion assessment from CTA as functional measures to assess the severity of the stenosis has conferred CTA with the opportunity to be the investigation of choice for the detection of patients with suspected graft dysfunction [75]. The authors believed that for graft assessment, CTA, with further improvement in scanning technologies, would supersede ICA in the near future [75]. A study by Weustink et al. [55] concluded that CTA had 100% diagnostic accuracy for the detection or exclusion of significant stenosis in grafts in symptomatic post-CABG patients. Specificity, sensitivity, PPV, and NPV were 100% each for the detection of significant stenosis. They believed CTA was very successful and had high diagnostic accuracy for the diagnosis of post-CABG significant stenosis or occlusion of the graft. It should be considered complementary, rather than a substitute to ICA in graft patency and stenosis detection and exclusion in symptomatic patients, but ICA is still required for confirmation of CT assessment for native coronary artery obstruction [55]. Though CTA is a noninvasive tool for the assessment of bypass grafts and native coronaries, its limitations include its inability to detect stenosis or obstruction in highly calcified coronaries, overestimation of stenosis because of calcification, poor visualization of distal anastomosis, low-quality images because of artifacts from hemostatic metal clips, patients’ inability to hold longer breath, fast heart rate, atrial fibrillation, and residual coronary motion. Contrary to ICA, being only diagnostic and not therapeutic is another limitation of CTA [54,55,62,75]. Radiation exposure is still a matter of concern. However, the recent generation whole-heart coverage CT scanner has addressed this issue to some extent. A study by Mushtaq et al. showed that CTA was able to interpret 100% of the bypass graft. CTA, in comparison with ICA, was able to detect occlusion and significant stenosis accurately in all CABG segments. The sensitivity, specificity, PPV, and NPV for the bypass graft were 100% each. They concluded that novel whole-heart coverage CT scanners had the assessment and evaluation capacity of bypass grafts with excellent interpretability, a lower level of radiation, and even in the presence of a fast heart rate [54].

### 3.3. Magnetic Resonance Angiography (MRA)

Magnetic resonance angiography (MRA) is another alternative for assessing the patency and quality of coronary bypass grafts. As invasive coronary angiography was associated with few minor complications, a noninvasive technique (MRA) that could assess the bypass graft was a major advancement in the field of medicine [78]. As a noninvasive alternative to ICA, multiple magnetic resonance (MR) techniques have been explored for graft evaluation and have been used clinically [79,80,81]. MR provides the best three-dimensional (3D) information with contrast and acquisition of graft images from any anatomical plane. In the beginning, magnetic resonance imaging (MRI) of the bypass grafts was obtained by non-respiratory compensation, spin echo, ECG triggered, and gradient echo techniques. Most studies are focused on classical spin echo and gradient echo techniques, which show that the results of these techniques are very good in detecting proximal graft patency but are limited in assessing different segments of sequential grafting and significant stenosis of the graft [82,83]. However, some studies suggest that MR flow measurement techniques may differentiate between the patent graft and significant stenosis of the graft [84]. Breath-hold 2D, 3D, and navigator-gated techniques further technically improved the functional and morphological assessment of the graft [78,83]. A study by Susan et al. concluded that navigator-gated 3D magnetic resonance angiography may be the best way to evaluate the graft and to achieve high-resolution volume images because the duration of breath hold did not affect image time [78]. Another study by Stauder et al. [66] assessed coronary artery bypass graft patency and flow, both at rest and in stress, with magnetic resonance angiography and magnetic resonance flow measurement. They evaluated 45 patients with 86 grafts 5.5 years after bypass surgery. There were 40 venous and 46 arterial grafts. The patency was assessed by MRA. The flow was checked by flow measurement at rest and also in stress induced by dipyridamole. Stenosis of all grafts’ segments was assessed by MRA and grafts were additionally seen by MRA and flow measurements. Invasive coronary angiography or multidetector computed tomography was used as a reference. Sensitivity, specificity, PPV, and NPV for stenosis were 100%, 97.8%, 87.5%, and 100% for venous grafts, and 95.2%, 96.8%, 80%, and 99.4% for arterial grafts, respectively [66]. In total, 84 out of 86 grafts (97%) were correctly classified using combined MRA and MR flow measurements. They concluded that magnetic resonance allowed the evaluation of both graft patency and flow in symptomatic patients after coronary artery bypass surgery [66]. Another angiographically controlled MRI study for postoperative graft patency assessment by Galjee et al. showed 96% predictive accuracy for venous grafts [67].

### 3.4. Transthoracic Doppler Echocardiography (TDE)

Transthoracic Doppler echocardiography (TDE) was first introduced by Gould in 1972 as a non-invasive tool for the assessment of venous grafts [85]. TDE has the potential for graft assessment in all postoperative patients owing to its lower risk, versatility, availability, and lower cost.

Since the 1990s, flow in the LIMA has been investigated using TDE [86,87]. Several studies have reported the usefulness of TDE for the LITA-LAD graft assessment [88,89,90,91,92,93]. A study by Hirata et al. examined 35 patients who underwent CABG using LITA-LAD grafting followed by a postoperative coronary angiographic examination [87]. By using the pulsed Doppler method, they measured the diameter and flow velocity of LITA, LAD, and the site of anastomosis using intraluminal flow signals. Under the guidance of color Doppler flow mapping, spectral Doppler recordings at the anastomosis site between LITA and LAD were obtained in 31 (89%) out of 35 patients [87]. In conclusion, they mentioned that noninvasive TDE compared favorably with coronary angiography and was considered to be reliable and able to provide greater information regarding the physiologic state in the area of anastomosis [87]. Additionally, a study by Driever et al. concluded that transcutaneous Doppler ultrasound was useful in the detection of LIMA graft flow and hence the results favored the routine use of this non-invasive technique in postoperative follow-up of patients with LIMA grafts [91]. Similarly, Van Son et al. concluded that Doppler spectrum analysis may be used in the postoperative serial assessment of IMA graft function [92]. Moreover, Yoshitatsu et al. aimed to investigate the changes in velocity profiles in the LAD after CABG by using TDE [94]. The Doppler velocity profile of the distal LAD was recorded with TDE in 45 patients before CABG and postoperatively after three weeks. As a result, they showed that TDE might be a noninvasive method to evaluate the effect of bypass grafting on the LAD. Furthermore, hemodynamics in bypass graft and recipient LAD artery was evaluated with combined 2D and TDE in 15 patients with IMA graft, and 24 patients with a saphenous vein graft. The graft vessel was detected in 11 of 14 (79%) patients with an IMA graft, and 20 of 23 (87%) patients with a saphenous vein graft. In conclusion, the Doppler method not only evaluated the direct effect of bypass grafting on coronary circulation but also the differences in effects between these two different grafting techniques [89].

Given the aforementioned studies, being a non-invasive, feasible, and versatile tool with a low risk of complications, TDE may be taken into consideration for the detection of IMA grafts. The ability of immediate, bedside, and late postoperative period assessment by TDE makes it a promising tool for assessments of IMA grafts, but highly skilled operators in performing and interpreting results are required.

## 4. Future Perspective

Taking ICA as a reference, the new intraoperative graft assessment modalities have their limitations. Even though they may help to identify failed grafts, the low sensitivity, specificity, PPV, and NPV, as shown in Table 1, indicate that it is hard to rely solely on the result of these modalities. We believe the experience and clinical judgment of the surgeon play a very important role when deciding about graft revision. A less experienced surgeon who depends only on the results of these technologies may end up overlooking many failed grafts which need immediate revision, and may also revise many unnecessary grafts, which, in turn, increase the risk of complication, morbidity, and mortality, as well as the cost of the surgery. Some authors believe that the difference in the results of IFI/TTFM and ICA assessment can be partly due to the kinking of the relatively longer bypassed grafts during chest closure, because IFI and TTFM are performed before chest closure while ICA is performed after chest closure.

We always evaluate the performance of these newer noninvasive technologies in comparison with ICA, but the reliability of ICA, especially in the case of postoperative graft assessment, will only be proved with the development of a technology that is better than ICA. A study including 45 patients with 57 grafts by Hol et al. [7] compared intraoperative angiography results with 3- and 12-month follow-up angiography. The sensitivity, specificity, PPV, and NPV of intraoperative angiography compared with follow-up angiography in their study were 42%, 82%, 38%, and 84%, respectively. They encountered 11 significant lesions using intraoperative angiography. Out of these eleven grafts, one vein graft which had 90% stenosis at anastomosis was revised. On follow-up angiography, 8/11 (73%) lesions were normal. The authors believed that occasionally it is hard to interpret the intraoperative angiography results because not all the negative results are important for the future patency of the graft. With an NPV of 84%, they believed that on-table angiography optimal results predicted good follow-up outcomes while the significant lesions had less impact because of the PPV of 38%.

Based on the intra- and postoperative noninvasive modalities result, as well as the result of ICA in this given study, it seems that there is still room for improvement in developing an easy-to-use, noninvasive graft assessment modalities that can provide both morphological and functional assessment of the graft, anastomosis, as well as the native coronaries, with no or minimum contrast and radiation. A high accuracy (specificity and sensitivity) would be the optimal goal of this technology so that a surgeon can easily rely on it.

## 5. Conclusions

In the quest for assessing the quality of intra- and postoperative coronary artery bypass grafts, ICA remains the gold standard modality. Despite its undeniable strengths, the limitations of ICA sometimes render it unfeasible for use during or after surgery. Fortunately, research suggests that IFI/ICG angiography is another viable option for intraoperative graft assessment followed by TTFM, while CTA emerges as the most reliable option for postoperative graft evaluation after ICA. However, it is important to note that relying solely on these modalities may lead to overlooking failed grafts that require immediate revision or performing unnecessary revisions. Therefore, a surgeon’s clinical judgment remains a crucial factor in the decision-making process.

## Figures and Tables

**Table 1 diagnostics-13-01891-t001:** Comparison of intraoperative noninvasive graft assessment modalities with standard invasive coronary angiography.

Author (Year of Publication)	Modality	No. of Patients	No. of Grafts	Types of Graft	Types of Surgery	Revision Rate	Specificity	Sensitivity	PPV	NPV
Desai et al. (2006) [23]	IFI	46	139	Mixed	ONCAB	3.6%	100%	83%	100%	98.40%
Desai et al. (2005) [24]	IFI	6	22	Mixed	Mixed	4.2%	100%	100%	100%	100%
# Walker et al. (2013) [25]	TTFM	160	160	Arterial	OPCAB	0%	94.1%	0%	0%	94.7%
Desai et al. (2006) [23]	TTFM	46	139	Mixed	ONCAB	3.6%	98.40%	25%	60%	93.20%
Hol et al. (2007) [26]	TTFM	39	39	Arterial	OPCAB	0%	100%	0%	0%	77%
Hol et al. (2007) [26]	HEE	39	39	Arterial	OPCAB	0%	90%	22%	40%	79%

#: In this study, the false positive refers to the graft with at least a single questionable parameter on the TTFM and found patent on angiography; ICA, invasive coronary angiography; TTFM, transient time flow measurement; HEE, high-frequency echocardiography; IFI, intraoperative fluorescence imaging; OPCAB, off-pump coronary artery bypass grafting; ONCAB, on-pump coronary artery bypass grafting; PPV, positive predictive value; NPV, negative predictive value.

**Table 2 diagnostics-13-01891-t002:** Comparison of postoperative noninvasive graft assessment modalities with standard invasive coronary angiography.

Author (Year of Publication)	Modality	No of Patients	No of Grafts	Types of Graft	Follow-Up (Years)	Specificity	Sensitivity	PPV	NPV
Mushtaq et al. (2019) [54]	CTA	100	283	Mixed	NA	100%	100%	100%	100%
Weustink et al. (2009) [55]	CTA	52	88	Mixed	9.6 ± 7.2	100%	100%	100%	100%
Turkvatan et al. (2009) [56]	CTA16 slice	102	236	Mixed	5.9	98.5%	91.4%	84.2%	99.2%
# Jabara et al. (2007) [57]	CTA 64 slice	50	147	Mixed	NA	100%	93.3%	100%	98.3%
						100%	100%	100%	100%
Houslay et al. (2007) [58]	CTA 16 slice	50	116	Mixed	7 ± 5	100%	92.8%	100%	85.8%
Malhothra et al. (2007) [59]	CTA 16 slice	114	338	Mixed	1.4 ± 0.4	100%	90.6%	100%	99%
Meyer et al. (2007) [60]	CTA 64 slice	138	418	Mixed	8.0 ± 5.2	97%	97%	93%	99%
* Anders et al. (2006) [61]	CTA 16 slice	32	94	Mixed	NA	95%	75%	53%	98%
						95%	69%	56%	97%
Malagutti et al. (2006) [62]	CTA 64 slice	52	109	Mixed	10 ± 5	98.3%	100%	98%	100%
Malhothra et al. (2005) [63]	CTA 16 slice	69	209	Mixed	1.3 ± 0.3	100%	81.8%	100%	99%
Schlosser et al. (2004) [64]	CTA 16 slice	48	131	Mixed	5.6 ± 6.2	95%	96%	81%	99%
Yoo et al. (2003) [65]	CTA	42	125	Mixed	Within 3 months	98.3%	100%	71.4%	100%
Stauder et al. (2007) [66]	MRA	46	46	Arterial	5.5	96.8%	95.2%	80%	99.4%
			40	Venous	5.5	97.8%	100%	87.5%	100%
Galjee et al. (1996) [67]	MRA SE	47	98	Venous	4.4 ± 3.9	85%	98%	97%	92%
Galjee et al. (1996) [67]	MRA Cine GE	47	98	Venous	4.4 ± 3.9	88%	98%	97%	94%
Lisboa et al. (2002) [68]	TDE	72	72	Arterial	Before discharge	96.8%	50%	NA	NA

# The first row in this study is the result of occluded grafts while the second row is of significant stenosed grafts (stenosis > 50%); * the first row in this study is the result of the first observer while the second row is the result of the second observer; CTA, computed tomographic angiography; MRA, magnetic resonance angiography; PPV, positive predictive value; NPV, negative predictive value; SE, spin echo; GE, gradient echo; TDE, transthoracic doppler echocardiography; NA, not available.

## Data Availability

All the data collected and analyzed during this study are included in this article.

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
