# Peer review of "Assessment of the Graft Quality and Patency during and after Coronary Artery Bypass Grafting"

_diagnostics, 2023, doi:10.3390/diagnostics13111891_

Round 1

Reviewer 1 Report

The manuscript "Assessment of the graft quality and patency during and after coronary artery bypass grafting"  by Masroor and coworkers intends to review the current modalities for graft assessment after CABG.

My comments are the following:

  1. Table 1 lists three papers by Desai and coworkers. Surely there are other groups that have looked into the matter
  2. The same Table 1, lists the results of Hol et al. These are copied as it is in the original manuscript, where they presented their result as fractions and not percentages. This is misleading and wrong. It it repeated in Lines 202-212.
  3. In Lines 226-234 they quote the research by Kuroda et al. It is unclear at which site they used flow-velocity measurement by TEE

Author Response

Response to reviewer 1: Respected Sir, we are very much grateful to you for giving your valuable time and reviewing this manuscript and we are sure your precious comments will further improve the quality of this manuscript. Please feel free to let us know if there is anything that needs further changes.

The manuscript "Assessment of the graft quality and patency during and after coronary artery bypass grafting"  by Masroor and coworkers intends to review the current modalities for graft assessment after CABG.

My comments are the following:

  1. Table 1 lists three papers by Desai and coworkers. Surely there are other groups that have looked into the matter

      Response: Thank you for the comment. Actually, there are lots of work available on intraoperative graft assessment using different modalities. But to our knowledge by using PUBMED, we only found the given studies that compared these comparatively new modalities with conventional invasive coronary angiography. So, we only included studies that have done comparisons with ICA. We would be grateful for letting us know if there is any in your knowledge that we have missed.

  1. The same Table 1, lists the results of Hol et al. These are copied as it is in the original manuscript, where they presented their result as fractions and not percentages. This is misleading and wrong. It is repeated in Lines 202-212.

Response: Thank you for the comment. For consistency and to avoid misunderstanding, the results of that specific study by Hol et al. have been converted from fractions to percentages both in Table 1 as well as in Lines 202-212.

  1. In Lines 226-234 they quote the research by Kuroda et al. It is unclear at which site they used flow-velocity measurement by TEE

Response: Thank you for the comment. The specific site of the LIMA at which the flow velocity was measured is not available in the original article cited by us. But it is understandable that it is very hard to locate LIMA at TEE. In the given study as soon as the researcher located the LIMA on TEE, the surgeon clamped the LIMA graft, and flow was measured in the graft with the help of TEE irrespective of the graft site.

Reviewer 2 Report

Thank you for asking me to review this manuscript.

Interesting work, well presented and would be of interest.

I only have few minor comments

- Can the authors ensure that abberviations are spelled the first time they appear in the text. On page 2 line 68, ICA needs to be spelled. This happens after line 70. Similarly, in table 1, HEE (the word epicardial is missing) to allow the reader to understand. Line 111, please spell TTFM, page 132, spell OR.

- On page 3, line 89, The hybrid interventional rooms are already established and in use rather than in development. Can the authors rephrase.

- Can the authors explain and elaborate what FitzGibbon types are. Not all readers will be famliar with this classifications.

- A minor comment about consistency, line 149 & 150, numbers are written rather than numerical like the rest of the manuscript.

- Line 165, what does middle anastomoses mean. Authors have to anticipate that not every reader is familiar with surgical terminology and their meanings.

- More recent references are needed. For instance, LIMA occlusion rate was up to 8.8% at 1 year in the latest publication in JACC Intervention. Please update and any relevant recent references in all aspects not just LIMA patency.

- line 249, the authors list echocardiography as a tool for graft quality assessment. Can the authors elaborate and explain how? It may be that they mean by LV impairment but that is not by any mean a proper indicator. 

- I am not sue coronary angiography is a complicated procedure, as stated on page 7, line 263,  and the statement that (line 302)"has conferred CTA superiority over ICA,  am not sure this is accurate and can be used. Please re-quantify your statement

- Line 317, arterial should be Atrial

All in all, an interesting subject that is presented well and would be of interest with the mentioned comments accommodated.

Author Response

Response to reviewer 2: Respected Sir, we are very much grateful to you for giving your precious time and reviewing this manuscript and we believe your precious comments will further improve the quality of this manuscript. Please feel free to let us know if there are any further changes needed.

Thank you for asking me to review this manuscript.

Interesting work, well presented and would be of interest.

I only have few minor comments

- Can the authors ensure that abberviations are spelled the first time they appear in the text. On page 2 line 68, ICA needs to be spelled. This happens after line 70. Similarly, in table 1, HEE (the word epicardial is missing) to allow the reader to understand. Line 111, please spell TTFM, page 132, spell OR.

Response: Thank you for the comment: All the corrections have been made except OR which was spelled on its first appearance in line 90.

- On page 3, line 89, The hybrid interventional rooms are already established and in use rather than in development. Can the authors rephrase.

Response: Thank you for the comment. What we mean in that sentence is the availability of hybrid suites in more hospitals which is not the case recently. We have rephrased the sentence.

- Can the authors explain and elaborate what FitzGibbon types are. Not all readers will be famliar with this classifications.

Response: Thank you for the comment: FitzGibbon classification has been explained.

- A minor comment about consistency, line 149 & 150, numbers are written rather than numerical like the rest of the manuscript.

Response: Thank you for the comment: Correction made as requested.

- Line 165, what does middle anastomoses mean. Authors have to anticipate that not every reader is familiar with surgical terminology and their meanings.

Response: Thank you for the comment: Correction and clarification made as requested.

- More recent references are needed. For instance, LIMA occlusion rate was up to 8.8% at 1 year in the latest publication in JACC Intervention. Please update and any relevant recent references in all aspects not just LIMA patency.

Response: Thank you for the comment: we have tried to insert new references if available and where possible.

- line 249, the authors list echocardiography as a tool for graft quality assessment. Can the authors elaborate and explain how? It may be that they mean by LV impairment but that is not by any mean a proper indicator. 

Response: Thank you for the comment. No, we did not mean the LV impairment assessment by echocardiography defines its role as a graft assessment tool. The use of echocardiography for graft quality and patency assessment is available in the literature. Even though it is not the best modality for graft quality assessment, researchers have achieved some good results while using it for graft patency which has been mentioned in the manuscript. We have specified a section for the role of echocardiography in the assessment of graft patency both in the intra- and postoperative periods in the manusript.

- I am not sue coronary angiography is a complicated procedure, as stated on page 7, line 263,  and the statement that (line 302)"has conferred CTA superiority over ICA,  am not sure this is accurate and can be used. Please re-quantify your statement

Response: Thank you for the comment. It is sure that ICA is an invasive procedure that is associated with some complications. Meanwhile, it is time-consuming and needs a team of experts to perform it, etc. We have mentioned all these limitations in the manuscript. Contrary to it, other non-invasive techniques are much simple and can be performed very easily. Considering this aspect of ICA we have said it is a complicated procedure but for better understanding we have changed the word “complicated” to “complex”. We hope that will clear the misconception.

Though the new CT scanners have improved a lot and overcome many limitations, still CTA has its own limitation which is discussed in the manuscript. Same as you, we also believe ICA is the gold standard for graft assessment both during the intra- and postoperative periods. We have mentioned this in the relevant sections of the manuscript as well as in the conclusion. The claim that new advancements in CTA have conferred superiority over ICA is the claim of that cited original article and we have mentioned that the authors believe so. As a review article, we have tried to include different opinions and as much literature as possible and tried to reflect the thoughts of different researchers.

- Line 317, arterial should be Atrial

Response: Thank you for the comment. Correction made as requested.

All in all, an interesting subject that is presented well and would be of interest with the mentioned comments accommodated.

Round 2

Reviewer 1 Report

The authors have not responded to the issues I raised. Specifically:

  1. Table 1 lists three papers by Desai and coworkers. Surely there are other groups that have looked into the matter

Response: Thank you for the comment. Actually, there are lots of work available on intraoperative graft assessment using different modalities. But to our knowledge by using PUBMED, we only found the given studies that compared these comparatively new modalities with conventional invasive coronary angiography. So, we only included studies that have done comparisons with ICA. We would be grateful for letting us know if there is any in your knowledge that we have missed.

Here are some additional papers that the authors should have included in Table 1. Some of these were published in high-impact journals and are newer than those included in Table 1. Interestingly, some of these appear in the Bibliography of the manuscript, but are not included in this Table. For a Review Paper, I consider that the purpose of the paper is not met.

Walker PF, Daniel WT, Moss E, Thourani VH, Kilgo P, Liberman HA, Devireddy C, Guyton RA, Puskas JD, Halkos ME. The accuracy of transit time flow measurement in predicting graft patency after coronary artery bypass grafting. Innovations (Phila). 2013 Nov-Dec;8(6):416-9. doi: 10.1097/IMI.0000000000000021. PMID: 24356431.

Jokinen JJ, Werkkala K, Vainikka T, Peräkylä T, Simpanen J, Ihlberg L. Clinical value of intra-operative transit-time flow measurement for coronary artery bypass grafting: a prospective angiography-controlled study. Eur J Cardiothorac Surg. 2011 Jun;39(6):918-23. doi: 10.1016/j.ejcts.2010.10.006. Epub 2010 Nov 20. PMID: 21095134. 

Waseda K, Ako J, Hasegawa T, Shimada Y, Ikeno F, Ishikawa T, Demura Y, Hatada K, Yock PG, Honda Y, Fitzgerald PJ, Takahashi M. Intraoperative fluorescence imaging system for on-site assessment of off-pump coronary artery bypass graft. JACC Cardiovasc Imaging. 2009 May;2(5):604-12. doi: 10.1016/j.jcmg.2008.12.028. PMID: 19442948.

Honda K, Okamura Y, Nishimura Y, Uchita S, Yuzaki M, Kaneko M, Yamamoto N, Kubo T, Akasaka T. Graft flow assessment using a transit time flow meter in fractional flow reserve-guided coronary artery bypass surgery. J Thorac Cardiovasc Surg. 2015 Jun;149(6):1622-8. doi: 10.1016/j.jtcvs.2015.02.050. Epub 2015 Feb 28. PMID: 25840755.

Singh SK, Desai ND, Chikazawa G, Tsuneyoshi H, Vincent J, Zagorski BM, Pen V, Moussa F, Cohen GN, Christakis GT, Fremes SE. The Graft Imaging to Improve Patency (GRIIP) clinical trial results. J Thorac Cardiovasc Surg. 2010 Feb;139(2):294-301, 301.e1. doi: 10.1016/j.jtcvs.2009.09.048. Epub 2009 Dec 16. PMID: 20006356. 

I still consider that Table 1 that includes 4 papers, of which 3 are from the same author should be revised, as well as the relevant paragraphs from the text. 

Round 3

Reviewer 1 Report

I’ve read the author’s response to the issues I raised and I consider it satisfactory and the explanations sensible. However, I would still refer the final decision to an Editor.